# Vitamin D Status Modestly Regulates NOD-Like Receptor Family with a Pyrin Domain 3 Inflammasome and Interleukin Profiles among Arab Adults

**DOI:** 10.3390/ijms242216377

**Published:** 2023-11-15

**Authors:** Sobhy M. Yakout, Hend Alfadul, Mohammed G. A. Ansari, Malak N. K. Khattak, Nasser M. Al-Daghri

**Affiliations:** Chair for Biomarkers of Chronic Diseases, Biochemistry Department, College of Science, King Saud University, Riyadh 11451, Saudi Arabiamkhattak@ksu.edu.sa (M.N.K.K.)

**Keywords:** circulating inflammasome, NLRP3, vitamin D deficiency, Saudi adults, inflammation, immune response

## Abstract

Vitamin D (VD) deficiency has been associated with inflammation and dysregulation of the immune system. The NLRP3 inflammasome, a critical immune response component, plays a pivotal role in developing inflammatory diseases. VD hinders NLRP3 inflammasome activation and thus exerts anti-inflammatory effects. This study aimed to analyze the effect of VD deficiency on circulating levels of NLRP3 inflammasomes (NLRP3 and caspase–1) and associated interleukins (IL–1α, IL–1β, IL–18, IL–33 and IL–37) in Saudi adults. Methods: A total of 338 Saudi adults (128 males and 210 females) (mean age = 41.2 ± 9.1 years and mean BMI 31.2 ± 6.5 kg/m^2^) were included. Overnight-fasting serum samples were collected. Participants were stratified according to their VD status. Serum levels of NLRP3 inflammasomes and interleukins of interest were assessed using commercially available immuno-assays. Individuals with VD deficiency had significantly lower mean 25(OH)D levels than those with a normal VD status (29.3 nmol/L vs. 74.2 nmol/L, *p* < 0.001). The NLRP3 levels were higher in the VD-deficient group than their VD-sufficient counterparts (0.18 vs. 0.16, *p* = 0.01). Significant inverse associations were observed between NLRP3 levels with age (r = −0.20, *p* = 0.003) and BMI (r = −0.17, *p* = 0.01). Stepwise regression analysis identified insulin (β = 0.38, *p* = 0.005) and NLRP3 (β = −1.33, *p* = 0.03) as significant predictors of VD status, explaining 18.3% of the variance. The findings suggest that the VD status modestly regulates NLRP3 inflammasome and interleukin activities. This may provide novel insights into the pathogenesis and management of inflammatory disorders.

## 1. Introduction

Vitamin D (VD) deficiency is a global health concern affecting different populations worldwide, including the Kingdom of Saudi Arabia (KSA). The prevalence of VD deficiency in the KSA is alarmingly high, with estimates ranging from 50% to 90%, depending on the region and age group studied [1,2,3,4,5]. VD deficiency is associated with an increased risk of various chronic diseases, including cardiovascular diseases, diabetes, cancer, and autoimmune disorders [6,7,8,9,10]. In recent years, there has been growing interest in the association between VD deficiency and inflammation [11,12,13,14,15].

VD, also known as the sunshine vitamin, was formerly thought to be involved only in maintaining calcium homeostasis, bone health, and skeletal development [11]. However, emerging evidence suggests that its functions extend beyond its well-established role in bone health [16,17,18,19]. VD has been implicated in regulating immune responses, including the modulation of pro-inflammatory cytokines and immune cell differentiation [20,21,22]. VD exerts a protective role by suppressing reactive oxygen species (ROS) generation and thus apoptosis and inflammation [23].

Inflammation is a complex biological response to harmful stimuli, such as pathogens or tissue injury. It is a tightly regulated process involving the activation of the immune system and the release of pro-inflammatory cytokines. However, dysregulated or chronic inflammation can contribute to the pathogenesis of several diseases, including cardiovascular diseases, metabolic disorders, and autoimmune conditions [24,25]. VD receptors (VDRs) play an anti-inflammatory role in both the innate and adaptive immunity [19]. VDRs are found in various tissues, including immune cells, indicating an immunomodulatory role [20].

Several research investigations have demonstrated the influential role of VD in moderating inflammation via the inflammasome pathway [26,27,28]. When VD interacts with its receptor, it acts to prevent the formation of the inflammasome complex and reduces the heightened inflammatory reactions leading to pathological conditions [26,27,28]. In a 2014 study we spearheaded, we assessed genetic interactions to understand the intricate link between the VDR structure and inflammation. The outcomes indicated that a particular variation in the VDR gene—a single nucleotide polymorphism—is closely linked to increased activation of the inflammasome complex and elevated concentrations of certain interleukins, such as IL–1β and IL–18 [29].

Inflammasomes serve as an essential element within the innate immune system, governing inflammatory responses [30,31,32]. The NLRP3 inflammasome, part of the NOD-like receptor family with a pyrin domain, stands out in recent research. Its association with numerous inflammatory-related conditions, such as diabetes [33] and Alzheimer’s disease [34], has heightened its significance. The NLRP3 inflammasome is a multiprotein complex that senses various danger signals and activates the pro-inflammatory cytokines ILs, such as IL–1β and IL–18 [30,31,32], amplifying the inflammatory response [35,36]. Dysregulated chronic hyperactivation of the NLRP3 inflammasome complex is linked to disease development. VD, a chief immune system regulator via its receptor, regulates NLRP3 inflammasome complex activity [37]. Rao et al. found that VDRs could directly interact with the NLRP3 protein [28].

Understanding the interplay between VD deficiency, NLRP3 inflammasome activation, and inflammation may provide valuable insights into the underlying mechanisms and potential therapeutic targets for managing inflammation-related diseases. We hypothesize that VD can suppress NLRP3 inflammasome activation and exert anti-inflammatory effects. Thus, we aim to analyze for the first time the impact of VD deficiency on circulating levels of NLRP3 inflammasome molecules (NLRP3 and caspase–1) and associated interleukins (IL–1α, IL–1β, IL–18, IL–33 and IL–37) in Saudi adults, as well as assess the correlations between VD and NLRP3 inflammasome molecules and the same ILs. 

## 2. Results

A total of 338 Saudi adults (128 males and 210 females, mean age 41.2 ± 9.1 years, mean BMI 31.2 ± 6.5 kg/m^2^) were included in the present study. Table 1 shows the clinical characteristics of age-, sex- and BMI-matched participants according to VD status. After adjusting for age and BMI, VD-sufficient participants had significantly lower total cholesterol levels than the VD-deficient group (*p* = 0.03). All other anthropometrics, lipids, and trace minerals differed significantly (*p* > 0.05) between groups.

Table 1 presents the clinical characteristics of 338 Saudi adults (128 males and 210 females, mean age 41.2 ± 9.1 years, mean BMI 31.2 ± 6.5 kg/m^2^) categorized by their VD status. A standout observation from the table is the pronounced difference in 25(OH)D levels across genders. Males with sufficient VD had median levels of 71.3 nmol/L, significantly higher than the deficient group’s level of 30.4 nmol/L (*p* < 0.001). Similarly, females in the sufficient category had levels of 75.7 nmol/L, contrasting sharply with 28.7 nmol/L in the deficient group (*p* < 0.001).

Age- and BMI-adjusted comparisons revealed that participants with adequate VD levels had notably lower total cholesterol levels than the VD-deficient group (*p* = 0.03). 

Other metrics, such as anthropometrics, lipids, and trace minerals, had no significant differences between groups.

Table 2 shows the circulating levels of NLRP3 inflammasome molecules and related interleukins. After adjusting for age and BMI, participants with VD deficiency had significantly lower levels of circulating NLRP3 than the VD-sufficient group (*p* = 0.05). All other NLRP3 inflammasome molecules, including caspase–1 and associated interleukins (IL–1α, IL–1β, IL–18, IL–33 and IL–37), did not show a significant difference (*p* > 0.05) between the VD-sufficient and VD-deficient groups. 

Correlation coefficients between NLRP3 levels and select parameters in the normal and VD-deficient groups are shown in Table 3. In the VD-sufficient group, NLRP3 levels were negatively but not significantly correlated with IL–18 levels (r = −0.12, *p* = 0.34). However, no significant correlations were observed between NLRP3 levels and the other parameters. Within the VD-deficient group, NLRP3 levels were inversely correlated with IL–33 levels (r = −0.20, *p* = 0.02). However, no significant correlations were found between NLRP3 levels and the remaining parameters. It is important to note that controlling for age and BMI in the analysis helped account for potential confounding factors that may influence the relationship between VD and NLRP3 levels. The observed negative correlation could be more confidently attributed to the specific relationship between VD and NLRP3 by controlling for these variables (Figure 1).

Table 4 presents the results of a stepwise regression analysis using VD as the dependent variable. The independent parameters in the analysis include age, BMI, WHR, blood pressure, glucose, HbA1c, insulin, total cholesterol, HDL cholesterol, triglycerides, NLRP3, caspase-1, IL–1α, IL–1β, IL–18, IL–33, and IL–37. Insulin positively correlated with VD levels (β = 0.05 ± 0.20, standardized β = 0.38, *p* = 0.005), indicating that higher insulin levels were associated with higher vitamin D levels. In contrast, NLRP3 demonstrated an inverse association with VD levels (β = −1.33 ± 0.59, standardized β = −0.29, *p* = 0.03), suggesting that higher NLRP3 levels were associated with lower VD levels. The stepwise regression model accounted for 18.3% of the variance in VD levels (adjusted r^2^ = 0.18). Age, BMI, log IL–18, log IL–1α, log IL–33, and log albumin were not retained in the final model as significant predictors.

## 3. Discussion

This study aimed to investigate the effect of VD on circulating levels of NLRP3 inflammasome molecules (NLRP3 and caspase-1) and associated interleukins (IL–1α, IL–1β, IL–18, IL–33, and IL–37) in Saudi adults. Our study revealed a significant inverse correlation between VD levels and NLRP3 expression. This finding is consistent with previous studies demonstrating the inhibitory effects of VD on the NLRP3 inflammasome activation [38,39]. 

VD has been shown to downregulate NLRP3 expression and caspase–1 activation, leading to decreased production of the pro-inflammatory cytokines IL–1β and IL–18 [38]. These findings suggest that VD deficiency may disrupt the balance of NLRP3-mediated inflammatory responses, leading to an exaggerated immune response and increased susceptibility to inflammatory diseases. VD has been recognized for its immunomodulatory properties, and its deficiency has been implicated in chronic inflammatory diseases [40]. Receptors for VD (VDRs) can be found in diverse immune cells such as macrophages, monocytes, and dendritic cells. Within these cells, they are instrumental in modulating immune reactions [41]. Activation of VDRs by VD metabolites can suppress pro-inflammatory cytokines and inhibit the production of reactive oxygen species, thereby exerting anti-inflammatory effects [22].

The NLRP3 inflammasome is a multiprotein complex that plays a central role in activating inflammatory responses [30,33]. It is involved in the maturation and release of pro-inflammatory cytokines, such as IL–1β and IL–18, and has been implicated in the pathogenesis of several inflammatory diseases [42]. Activation of the NLRP3 inflammasome can be triggered by various stimuli, including microbial components, damage-associated molecular patterns, and environmental factors [33,35,39]. Emerging evidence suggests that VD regulates NLRP3 inflammasome metabolism and function.

It is worth noting that our study also revealed significant associations between VD deficiency and other clinical parameters, including BMI, cholesterol levels, and IL–18 levels. These findings are consistent with previous studies demonstrating the impact of VD deficiency on metabolic factors and inflammatory markers [40]. VD has been shown to influence adipocyte function, lipid metabolism, and cholesterol synthesis [40]. Moreover, VD deficiency has been associated with increased IL–18 levels, consistent with our findings [43]. IL–18 is a pro-inflammatory cytokine implicated in the pathogenesis of various inflammatory diseases, including atherosclerosis and metabolic syndrome [44,45].

One possible mechanism underlying the association between VD deficiency and increased NLRP3 levels involves regulating VDR expression and activity. Studies have shown that VD deficiency can downregulate VDR expression, impairing signaling and reducing immune regulatory functions [20]. This dysregulation may disrupt the balance between pro-inflammatory and anti-inflammatory pathways, contributing to NLRP3 inflammasome activation and increased production of pro-inflammatory cytokines [46]. In addition, VD deficiency has been linked to oxidative stress and mitochondrial dysfunction, both of which can activate the NLRP3 inflammasome [40]. Another potential mechanism involves direct modulation of NLRP3 expression and activity by VD. Several studies have reported that VD can inhibit NLRP3 inflammasome activation and subsequent release of IL–1β and IL–18 [47,48]. VD has been shown to suppress NLRP3 expression, inhibit caspase–1 activation, and attenuate the production of pro-inflammatory cytokines [38,49]. These findings suggest that VD may exert its anti-inflammatory effects, at least in part, through regulating NLRP3 inflammasome activity.

The findings from this study offer valuable insights into the relationship between VD deficiency and NLRP3 inflammasome activation in Saudi adults. However, several limitations should be acknowledged. The cross-sectional design of the study limits our ability to establish causality or determine the temporal relationship between VD deficiency and NLRP3 activation. Longitudinal studies are necessary to confirm these findings and elucidate the direction of the association. Our study primarily focused on Saudi adults, limiting the generalizability of the results to other populations. It is essential for future research to encompass larger and more diverse cohorts to validate the observed associations. An oversight in our research was not accounting for potential seasonal variations in VD levels, which could introduce variability given that sunlight exposure influences VD synthesis. Moreover, factors such as dietary habits, genetic variations, and other environmental factors beyond sunlight exposure might influence both the VD status and NLRP3 levels. These potential confounders warrant further investigation in relation to their impact on our observations.

## 4. Materials and Methods

### 4.1. Study Design and Participants

We conducted a cross-sectional analysis using data from 338 Saudi adults, comprising 128 males and 210 females. The average age of the participants was 41.2 ± 9.1 years, with an average BMI of 31.2 ± 6.5 kg/m^2^. These data were sourced from the Chair for Biomarkers of Chronic Diseases (CBCD) main database at King Saud University (KSU) College of Science, Riyadh, Saudi Arabia. The CBCD, in collaboration with the Ministry of Health (MOH), maintains a comprehensive registry that holds clinical details and blood samples from over 10,000 Saudis aged between 1 and 99 years. These individuals were approached via various primary care facilities in Riyadh, Saudi Arabia [50,51].

The exclusion criteria considered for this study encompassed individuals with pre-existing cardiovascular disease, inflammatory systemic disease, eating disorders, celiac disease, inflammatory bowel disease, those with multi-organ complications, and patients who had been taking VD supplements or medications known to affect VD levels for at least 6 months prior to the study. Further exclusions were made for individuals who had undergone systemic glucocorticoid use (accumulating to one month of use in the year prior to study inclusion), non-consenting individuals, pregnant participants, and those aiming to conceive.

Before being included in the database, every participant provided their written consent (project # E-20–5127). The research received ethical clearance from the Research Center at the College of Science, KSU, Riyadh, Saudi Arabia (Ref 21/0412/IRB). Each participant filled out a questionnaire capturing demographic details, overall health, and previous medical records.

### 4.2. Sample Collection and Anthropometrics

Blood and serum samples from the chosen participants were sourced from the CBCD’s Biobank database at KSU. In essence, trained nurses and technicians at their respective PCCs collected overnight fasting blood and serum samples. For serum extraction, blood was allowed to clot without disruption in red-top tubes at 25 °C, ranging from 30 min up to an hour. Subsequently, these samples underwent centrifugation at 1000 to 1500× *g* for 10 min at 4 °C to separate the clot from the serum. Post-centrifugation, the isolated serum was partitioned, accurately labeled, and transported to CBCD at KSU for immediate storage at −80 °C until the scheduled analysis [52]. Experienced nurses recorded anthropometrics, encompassing weight, height, waist and hip measurements, and blood pressure readings. BMI and WHR were calculated using standard formulas: weight (in kg) divided by the square of height (in meters) and the ratio of waist to hip measurements, respectively. Based on the measured serum 25(OH)D levels, in line with national and regional directives [53], participants were grouped into VD-sufficient (≥50 nmol/L) or VD-deficient (<50 nmol/L) statuses.

### 4.3. Lipid Profile and Biochemical Estimations

Before biochemical evaluations, serum samples from chosen participants were thawed. Using the automated biochemical analyzer Konelab 20 from Thermo Fischer, Espoo, Finland, levels of serum triglycerides, total cholesterol, high-density lipoprotein (HDL) cholesterol, and glucose were determined [33]. HbA1c levels were directly assessed through ion-exchange high-performance liquid chromatography (HPLC) utilizing the D-10 Hemoglobin Testing System from Bio-Rad, Hercules, CA, USA. Insulin levels in fasting serum were gauged with the Luminex multiplex system from Luminex Corp, Austin, TX, USA. The intra-assay CV% was below 10 and the inter-assay CV% was under 15, according to the manufacturer’s guidelines. Using the COBAS e–411 automated analyzer from Roche Diagnostics, Indianapolis, IN, USA, and the Elecsys^®^ Vitamin D total II assay, serum 25(OH)D concentrations were ascertained. This assay boasts a spectrum measuring from 7.5 to 250 nmol/L, consistency under 6%, and mid-level precision below 7%, as indicated by the manufacturer [54].

### 4.4. Serum NLRP3 Inflammasome (NLRP3 and Caspase–1) and Interleukin (IL–1α, IL–1β, IL–18, IL–33 and IL–37) Estimations

Circulating NLRP3 levels were determined using ELISA kits from Cusabio, Houston, TX, USA (catalog number CSB–E15885h). Per the manufacturer’s guidelines, this assay has a detectable minimum value of <0.039 ng/mL for human NLRP3 and CV% of <8% and <10% for intra- and inter-assay precision, respectively [33]. The Flex MAP–3D System from Luminex Corporation, Austin, TX, USA was employed to measure circulating IL–1β, IL–18, and IL–37 levels using specific human cytokine magnetic bead panels. The CV% for IL–18, IL–1β, and IL–37 were: intra-assay <15, <20, and <10, and inter-assay <20, <15, and <15, respectively. IL–1α and IL–33 levels were determined using ELISA kits from Bio Vendor, R & D systems, Brno, Czech Republic (Cat No. RAF045R and RAF064R). Their intra-assay CV% was <5.4% and 4.7%, and inter-assay CV% was <10% and 6.9%. The CBCD’s quality assurance team at KSU consistently monitored the controls and standards used in these biochemical assays.

### 4.5. Statistical Analysis

Data were analyzed with SPSS (version 22, Chicago, IL, USA). Continuous variables with normal distributions were presented as means ± standard deviations (SD), while non-normal variables were shown as median (25th and 75th) percentiles. Categorical variables were expressed as percentages (%). Normality was tested using the Kolmogorov–Smirnov test. Non-normal variables were log-transformed prior to analysis. Independent sample *t*-test and the Mann—Whitney U test were performed to compare mean and median differences in normal and non-normal variables between genders and 25(OH) vitamin D statuses, and univariate analyses adjusted for age and BMI. Correlation analyses with NLRP3 and variables of interest were conducted using Pearson’s and Spearman’s analyses. Stepwise regression included log 25(OH)D as a dependent variable and age, BMI, log IL–18, log IL–α, log IL–33, log insulin, and log NLRP3 as independent variables to observe independent predictors for vitamin D. *P* < 0.05 was considered significant.

## 5. Conclusions

This study demonstrates a significant association between vitamin D deficiency and increased NLRP3 levels in Saudi adults. The negative correlation between vitamin D and NLRP3 levels suggests a potential role for vitamin D in modulating inflammasome activation. VD deficiency may modestly disrupt the regulatory mechanisms controlling NLRP3 inflammasome activation, leading to enhanced inflammation and increased susceptibility to inflammatory diseases. These findings highlight the importance of maintaining optimal VD levels for immune homeostasis and suggest potential therapeutic implications for targeting the NLRP3 inflammasome in the management of inflammatory disorders.

## Figures and Tables

**Figure 1 ijms-24-16377-f001:**
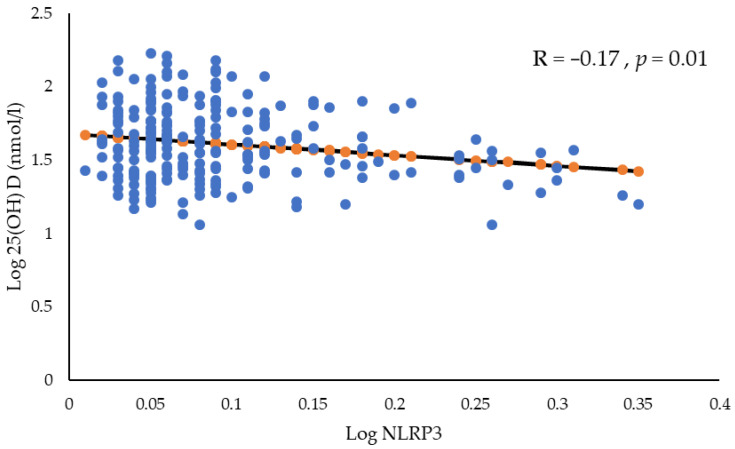
Relationship between VD and NLRP3 after adjusting for age and BMI.

**Table 1 ijms-24-16377-t001:** Clinical characteristics of participants according to VD status and gender.

Parameters	Males		*p*-Value	Females		*p*-Value
	Sufficient (>50 nmol/L)	Deficient (<50 nmol/L)		Sufficient (>50 nmol/L)	Deficient (<50 nmol/L)	
N	32	96		95	115	
Age (years)	42.9 ± 10.4	39.2 ± 9.8	0.07	42.7 ± 7.7	41.3 ± 8.8	0.11
BMI (kg/m^2^)	30.4 ± 6.6	29.0 ± 5.5	0.31	33.1 ± 6.6	31.8 ± 6.8	0.18
WHR	0.91 ± 0.1	0.92 ± 0.1	0.99	0.85 ± 0.1	0.85 ± 0.1	0.91
Systolic BP (mmHg)	123.1 ± 9.3	124.6 ± 13.9	0.59	126.1 ± 13.3	123.0 ± 14.8	0.13
Diastolic BP (mmHg)	76.8 ± 9.1	76.3 ± 10.3	0.53	78.8 ± 11.6	76.8 ± 10.6	0.19
Total Cholesterol (mmol/L)	4.7 ± 1.0	5.31 ± 1.0	0.008	5.03 ± 1.1	5.22 ± 1.1	0.22
HDL Cholesterol (mmol/L)	1.0 ± 0.2	1.02 ± 0.3	0.99	1.15 ± 0.3	1.14 ± 0.4	0.86
Triglycerides (mmol/L)	1.7 (1.3–2.2)	1.62 (1.1–2.3)	0.93	1.35 (1.11–2.0)	1.59 (1.1–2.1)	0.23
Insulin (uU/mL)	18.8 (16–20)	16.1 (11–17)	0.07	16.5 (15–19)	15.9 (11–19)	0.54
Glucose (mmol/L)	5.7 (5.4–6.3)	5.42 (4.9–6.2)	0.30	6.10 (5.3–7.4)	5.94 (5.2–7.5)	0.62
HbA1c (%)	5.84 ± 0.8	5.50 ± 0.9	0.07	5.75 ± 1.3	6.10 ± 1.5	0.08
Calcium (mmol/L)	2.21 ± 0.1	2.20 ± 0.2	0.68	2.21 ± 0.2	2.21 ± 0.2	0.89
25(OH)D (nmol/L)	71.3 (58–81)	30.4 (26–38)	<0.001	75.7 (62–94)	28.7 (22–41)	<0.001

Note: Data are presented as mean ± SD and median (25th–75th) percentiles; *p*-values were obtained from the independent sample *t*-test and Mann—Whitney U test. *p* < 0.05 is considered significant. WHR: Waist-to-Hip Ratio.

**Table 2 ijms-24-16377-t002:** Circulating levels of NLRP3 inflammasome molecules (NLRP3 and caspase–1) and associated interleukins (IL–1α, IL–1β, IL–18, IL–33 and IL–37).

Parameters	VD Status		*p*-Value	*p*-Value *
	Sufficient (≥50 nmol/L)	Deficient (<50 nmol/L)		
IL–1α (pg/mL)	0.6 (0.5–0.8)	0.6 (0.5–0.9)	0.84	0.91
IL–1β (pg/mL)	0.8 (0.7–3.0)	1.4 (0.7–2.9)	0.43	0.51
IL–18 (pg/mL)	9.2 (1.0–23.4)	10.5 (0.8–33.0)	0.69	0.92
IL–33 (pg/mL)	3.5 (3.0–4.0)	3.7 (3.2–4.0)	0.01	0.06
IL–37 (pg/mL)	2.9 (2.1–7.0)	4.2 (2.3–7.6)	0.53	0.49
Caspase–1 (ng/mL)	0.9 (0.5–2.7)	0.7 (0.5–2.1)	0.27	0.44
NLRP3 (ng/mL)	0.16 (0.12–0.24)	0.18 (0.11–0.32)	0.01	0.05

Note: *p*-values are obtained from the independent *t*-test; * denotes age- and BMI-adjusted *p*-values; *p* < 0.05 is considered significant.

**Table 3 ijms-24-16377-t003:** Correlations between NLRP3 and select parameters according to the VD status.

Parameters	Sufficient		Deficient	
	R	*p*-Value	R	*p*-Value
Age (years)	−0.12	0.28	−0.20	0.02
BMI (kg/m^2^)	−0.22	0.06	−0.13	0.12
WHR	0.09	0.48	−0.04	0.64
Systolic BP (mmHg)	0.00	0.99	−0.05	0.56
Diastolic BP (mmHg)	−0.03	0.77	−0.03	0.70
Cholesterol (mmol/L)	0.04	0.71	−0.03	0.69
Glucose (mmol/L)	0.05	0.64	−0.09	0.29
HDL Cholesterol (mmol/L)	0.03	0.77	−0.12	0.14
Triglycerides (mmol/L)	0.01	0.91	0.01	0.92
HbA1c	−0.02	0.86	−0.11	0.18
25(OH)D (nmol/L)	−0.10	0.39	−0.13	0.12
IL–18 (pg/mL)	−0.12	0.34	−0.26	0.02
IL–1α (pg/mL)	0.33	0.006	0.28	0.001
IL–β (pg/mL)	0.00	0.98	0.03	0.79
IL–33 (pg/mL)	0.37	0.001	0.20	0.02
IL–37 (pg/mL)	−0.04	0.72	0.10	0.34
Caspase–1 (ng/mL)	−0.11	0.40	0.02	0.83
Insulin (uU/mL)	−0.23	0.18	−0.25	0.08
Albumin (g/L)	−0.03	0.80	0.06	0.64
Calcium (mmol/L)	−0.33	0.01	0.06	0.65

Note: Data are presented as Pearson’s correlation coefficients (R); *p* < 0.05 is considered significant.

**Table 4 ijms-24-16377-t004:** Stepwise regression analysis.

Parameters	β ± SE	St. β	*p*-Value
Insulin	0.05 ± 0.2	0.38	0.005
NLRP3	−1.33 ± 0.59	−0.29	0.03
Adjusted R^2^	18.3		0.004

Note: Data are presented as Beta ± standard error obtained from the stepwise linear regression analysis; *p* < 0.05 is considered significant. Stepwise regression included log 25(OH)D as a dependent variable and age, BMI, log IL–18, log IL–α, log IL–33, log insulin, and logNLRP3 as independent variables.

## Data Availability

Data are available upon request from the corresponding author.

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
