# Peer review of "Vitamin D Status Modestly Regulates NOD-Like Receptor Family with a Pyrin Domain 3 Inflammasome and Interleukin Profiles among Arab Adults"

_ijms, 2023, doi:10.3390/ijms242216377_

Round 1

Reviewer 1 Report

Comments and Suggestions for Authors

It is known that Vitamin D plays an important role in modulating the immune and inflammatory systems by regulating the synthesis of inflammatory cytokines and inhibiting pro-inflammatory cell proliferation. Both of these processes are critical to the pathogenesis of inflammatory diseases. The work: “Associations of Vitamin D Status on NLRP3 Inflammasome and Interleukins Among Arab Adults”, is an original manuscript, fits these scientific trends.

Although the authors compiled a huge amount of data, I missed the gender breakdown of patients (into men and women, including pre- and post-menopausal). These characteristics have important implications on the evaluated parameters. We do not know if the patients were treated with any medications. The literature is poorly cited and needs improvement (there are missing citations, or their inappropriate order)

Author Response

It is known that Vitamin D plays an important role in modulating the immune and inflammatory systems by regulating the synthesis of inflammatory cytokines and inhibiting pro-inflammatory cell proliferation. Both of these processes are critical to the pathogenesis of inflammatory diseases. The work: “Associations of Vitamin D Status on NLRP3 Inflammasome and Interleukins Among Arab Adults”, is an original manuscript, fits these scientific trends.

  • Although the authors compiled a huge amount of data, I missed the gender breakdown of patients (into men and women). These characteristics have important implications on the evaluated parameters.

Thank you for your feedback. We'd like to point out that the gender breakdown of patients, segmented by their Vitamin D status, is provided in Table 1 of the revised manuscript. We recognize the importance of this distinction and have ensured its inclusion in our updated data presentation.

  • We do not know if the patients were treated with any medications.

Thank you for highlighting the oversight regarding the exclusion criteria. We have now updated the manuscript to specify the exclusion of patients taking vitamin D supplements or related medications for at least 6 months prior to the study.

  • The literature is poorly cited and needs improvement (there are missing citations, or their inappropriate order)

Thank you for your feedback regarding the citation inconsistencies in our manuscript. We have thoroughly reviewed the entire manuscript for citation accuracy and consistency. We have cross-checked all statements with their original sources, ensured the correct order of citations, and added any missing references. We've also had a colleague peer-review the revised manuscript with a focus on citations. We believe these efforts have substantially improved the citation process in our manuscript.

Reviewer 2 Report

Comments and Suggestions for Authors

Observations:

1. Table 1 row 97 - sufficient is >50nmol/L

2. WHR - is not described what is means and how is calculated

3. row 112 - correlation very week and not significant! I would say does not correlate

4. row 166 - statement should be totaly revverse. Vitamin D regulates inflammasome.

5. row 224 - centrifugation is to separate the clot from serum, not to remove it. Pay attention to language.

6. row 238, 244 - for Material and methods is not necessary to write the catalogue code. Is sufficient name of kit, producer and CV% (intra-, inter-) of the method

7. Major observation - vitamin D presents seasonal variations. Were these noticed? How do you interprete than the whole outcome?

Author Response

Observations:

  1. Table 1 row 97 - sufficient is >50nmol/L

Response

Thank you for pointing out the discrepancy in Table 1. We apologize for the oversight. It was a typographical error, and we have corrected it to reflect that "sufficient is >50nmol/L" in the revised manuscript. We appreciate your careful review and feedback.

  1. WHR - is not described what is means and how is calculated

The description and calculation of BMI and WHR were already provided in the Materials and Methods section on page 7, under the "Sample Collection and Anthropometrics" subsection.

  1. row 112 - correlation very week and not significant! I would say does not correlate

We have updated the wording to "negatively but not significantly correlated" in the revised manuscript to better reflect the findings.

  1. row 166 - statement should be totaly revverse. Vitamin D regulates inflammasome.

We have revised the statement to correctly reflect that "Vitamin D regulates inflammasome" in the updated manuscript.

  1. row 224 - centrifugation is to separate the clot from serum, not to remove it. Pay attention to language.

Thank you for highlighting this language error. We have made the necessary corrections in the revised manuscript to accurately describe the process.

  1. row 238, 244 - for Material and methods is not necessary to write the catalogue code. Is sufficient name of kit, producer and CV% (intra-, inter-) of the method

We have revised the manuscript to include only the name of the kit, the producer, and the CV% (intra-, inter-) of the method, omitting the catalogue code as advised.

  1. Major observation - vitamin D presents seasonal variations. Were these noticed? How do you interpret than the whole outcome?

Thank you for pointing out the oversight regarding seasonal variations in Vitamin D levels. We acknowledge this gap in our study. In response, we addressed this by adding it on limitations in the revised manuscript to highlight the potential impact of not considering seasonal fluctuations.

Round 2

Reviewer 1 Report

Comments and Suggestions for Authors

 Accept in present form

Author Response

We are extremely grateful to the reviewer for appreciating the substantial revisions made.

Reviewer 2 Report

Comments and Suggestions for Authors

The title is unclear. I strongly suggest to change it to a more general one, because the outcome of the study is not very strong. Correlations are weak and no clear statement can than be made.

Author Response

Comment: The title is unclear. I strongly suggest to change it to a more general one, because the outcome of the study is not very strong. Correlations are weak and no clear statement can than be made.

Response: We thank the reviewer for this comment and have changed the title accordingly to 'Vitamin D Status Modestly Regulates NLRP3 Inflammasome and Interleukin Profiles Among Arab Adults', since the main findings point to significant (but weak) inverse associations between vitamin D status,  circulating NLRP3 inflammasome and interleukins. We believe the revised title is more apt and reflective of the study findings.